# Evaluation of *Rouxiella badensis* Subsp *Acadiensis* (Canan SV-53) as a Potential Probiotic Bacterium

**DOI:** 10.3390/microorganisms11051347

**Published:** 2023-05-20

**Authors:** Ivanna Novotny-Nuñez, Gabriela Perdigón, Chantal Matar, María José Martínez Monteros, Nour Yahfoufi, Silvia Inés Cazorla, Carolina Maldonado-Galdeano

**Affiliations:** 1Laboratorio de Inmunología, Centro de Referencia para Lactobacilos (CERELA-CONICET), San Miguel de Tucumán T4000, Argentina; ivannanovotnyn@gmail.com (I.N.-N.); perdigon@cerela.org.ar (G.P.); mjmartinez@cerela.org.ar (M.J.M.M.); 2Department of Cellular and Molecular Medicine, Faculty of Medicine, University of Ottawa, Ottawa, ON K1H 8M5, Canada; chantal.matar@uottawa.ca (C.M.); nyahf074@uottawa.ca (N.Y.); 3School of Nutrition Sciences, Faculty of Health Sciences, University of Ottawa, Ottawa, ON K1N 6N5, Canada; 4Cátedra de Inmunología, Facultad de Bioquímica, Química y Farmacia, Universidad Nacional de Tucumán, San Miguel de Tucumán T4000, Argentina

**Keywords:** *R. acadiensis*, probiotic capacity, intestinal barrier

## Abstract

The advent of omic platforms revealed the significant benefits of probiotics in the prevention of many infectious diseases. This led to a growing interest in novel strains of probiotics endowed with health characteristics related to microbiome and immune modulation. Therefore, autochthonous bacteria in plant ecosystems might offer a good source for novel next-generation probiotics. The main objective of this study was to analyze the effect of *Rouxiella badensis acadiensis Canan* (*R. acadiensis)* a bacterium isolated from the blueberry biota, on the mammalian intestinal ecosystem and its potential as a probiotic microorganism. *R*. *acadiensis*, reinforced the intestinal epithelial barrier avoiding bacterial translocation from the gut to deep tissues, even after feeding BALB/c mice for a prolonged period of time. Moreover, diet supplementation with *R. acadiensis* led to increases in the number of Paneth cells, well as an increase in the antimicrobial peptide α defensin. The anti-bacterial effect of *R*. *acadiensis* against *Staphylococcus aureus* and *Salmonella enterica* serovar Typhimurium was also reported. Importantly, *R. acadiensis*-fed animals showed better survival in an in vivo *Salmonella enterica* serovar Typhimurium challenge compared with those that received a conventional diet. These results demonstrated that *R. acadiensis* possesses characteristics of a probiotic strain by contributing to the reinforcement and maintenance of intestinal homeostasis.

## 1. Introduction

The gut microbiota is a complex and active community of microorganisms, which comprises more than 100 billion microorganisms, including more than 1000 different bacterial species that play an important role in promoting health [1]. Commensal microbes in the intestine have numerous and important functions in the human body, including protection against pathogens colonization, maintenance of the gut mucosal immunity, and improvement of gut motility and they also play a vital role in digestion and metabolism. The influence of the resident or commensal microorganisms on mucosal immune function and gut health is an area of scientific and clinical importance [2]. The commensal microbes constantly coexist in a mutualistic relationship with the intestinal eukaryotic cells.

It is well established that there is an active dialogue between commensal bacteria and the host mucosal immune system. This crosstalk elicits differential responses from the immune system to commensal and pathogenic bacteria, through interaction with the same molecular patterns that are recognized by the Toll-like receptors family (TLR) [3,4].

Gastrointestinal cells are in symbiosis with microbial communities and endowed with mechanisms to recognize pathogens and microorganisms [5]. A healthy intestinal barrier plays an important role in the defense mechanisms. There is reciprocal communication between commensal microorganisms and the epithelial barrier cells, including epithelial, Goblet and Paneth cells [6,7]. This relationship contributes to maintaining the intestinal barrier by preventing the translocation of microorganisms to the tissues, and by preventing invasion by pathogenic microorganisms. Although the underlining mechanisms governing the interaction of epithelial cells with immune cells and pathogenic bacteria have been extensively studied, much less research has been conducted regarding the non-pathogenic microorganisms naturally present in foods. These microorganisms are called “probiotic bacteria” [8,9]. The “probiotic” concept was originally used to describe microbial feed supplements that stimulate the growth of farm animals [10]. Currently, the use of probiotics as dietary supplements has been largely extended to humans and could be used as a strategy to improve the intestinal barrier and the health of the host. 

Numerous health-related benefits have been claimed for probiotics, including the reinforcement of the gut barrier, improvement of the gut microbiome, prevention of certain infectious diseases, reduction of serum cholesterol, alleviation of allergy processes and inflammatory bowel disease (IBD) symptoms, as well as, anticarcinogenic activity and immune adjuvant properties [11,12,13,14,15,16]. The microorganisms most commonly used for probiotic product development are lactic acid bacteria, including species of lactobacilli, bifidobacteria, streptococci, lactococci, and some yeast strains (*Saccharomyces*). 

Currently, there is a growing interest in new available probiotic strains that would improve intestinal homeostasis in diseases such as diarrhea, obesity and inflammatory bowel disease (IBD). The selection of an appropriate probiotic microorganism should be made on the basis of its capacity to improve the gut ecosystem without disrupting intestinal homeostasis [17,18]. 

In order to be used as probiotics, microorganisms must meet certain minimum requirements: (a) be characterized at the genus, species and strain level; (b) be safe for the intended use [19,20]. The probiotic strain must not be hazardous to health. The bacterium must not present or promote bacterial translocation. In addition, the following complementary tests should be performed: resistance to antibiotics, hemolytic activity and toxin production; (c) in vitro tests that demonstrate the adjudicated effects (resistance to digestive enzymes, antimicrobial activity against potentially pathogenic bacteria; (d) characteristics that contribute to their colonization in the host: tolerance to low gastric pH, resistance to bile salts and adhesion to the host epithelium. Additionally, candidate microorganisms that do not permanently colonize the intestine, should demonstrate their ability to survive digestive tract harsh conditions after oral administration. Moreover, pre-clinical and clinical studies are also required in order to corroborate the health beneficial effects of candidate microorganisms [19,21].

Additionally, the functional dose required for the probiotic strain, as well as the effects and safety of their continuous or periodical consumption, must be known [22]. The in vitro sensitivity of *R. acadiensis* to antibiotics, and its tolerance to gastric and pancreatic simulated juices, were recently reported [23]. The main objective of this study was to further analyze the effects of Canan SV-53™, a proprietary strain of *Rouxiella badensis* subsp *acadiensis* (*R. acadiensis*), on the intestinal ecosystem after oral administration, and to determine its effects on the immune homeostasis, thus furthering the knowledge supporting its application as a potential probiotic microorganism. The behavior of the strain in the gut after different supplementation periods and its influence on the intestinal ecosystem were also investigated. Finally, *R. acadiensis* ability to protect against *Salmonella* Typhimurium infection and the immune response elicited was also explored. The results consolidated existing data about *R. acadiensis* as a probiotic microorganism with beneficial properties pertaining to human health.

## 2. Materials and Methods

### 2.1. Bacteria Strains

*Rouxiella badensis* subsp *acadiensis* (Canan SV-53), formerly known as *Serratia vaccini*. This bacterium has been filed in a U.S. Provisional Application No. 62/916,921 entitled “Probiotics Composition and Methods” for its potential probiotic effects, was provided by Chantal Matar from the University of Ottawa. The strain was isolated from blueberry microbiota [24,25]. 

The bacterium was grown in Tripticase Soy Agar (TSA) (Britania, Buenos Aires, Argentina) at 30 °C for 24 h. 

*Staphylococcus aureus* ATCC 25923 was kindly provided by Mariel Cáceres, Hospital Ángel C. Padilla, Tucumán. *Salmonella enterica* serovar Typhimurium strain was obtained from the Bacteriology Department of the Hospital del Niño Jesus (San Miguel de Tucumán, Argentina). 

Aliquots of *S.* Typhimurium and *S. aureus* (200 μL) from an overnight culture were placed in 5 mL of sterile Brain Heart Infusion (BHI) broth (Britania, Buenos Aires, Argentina) and incubated during 4–5 h to reach the exponential growth phase.

### 2.2. Animals and Diet Supplementation

Female six-week-old BALB/c mice (weight, 26 ± 4 g) were obtained from a closed random bred colony maintained at CERELA (Centro de Referencia para Lactobacilos, San Miguel de Tucumán, Tucumán, Argentina). Animals were kept in a controlled atmosphere (22 ± 2 °C; 55 ± 2% relative humidity) with a 12 h light/dark cycle, fed with commercially available conventional food and drinking water *ad libitum*.

*R. acadiensis* overnight cultures were grown at 30 °C in 5 mL of sterile TSA. The cells were harvested by centrifugation at 5000× *g* for 10 min, washed three times with phosphate-buffered saline solution (PBS) and resuspended in 5 mL of sterile 10% (wt/v) skim milk powder. Bacterial suspensions were diluted at 1:30 in water and administered to the mice. The final concentration of the bacteria was 2 ± 1 × 10^9^ CFU/mL. These counts were periodically controlled at the beginning of the administration and each 24 h of dilution in water to avoid modifications of more than one logarithmic unit. 

BALB/c mice were divided into groups of five animals each of them. One group (Normal Control) received a conventional diet and water *ad libitum* and the other group (test group) received a conventional diet plus 10^9^ CFU/mL of *R. acadiensis* in the drinking water for seven consecutive days. The choice of a 7-day time frame was extensively studied in our laboratory with other probiotic bacteria (lactobacilli) and determined to be the required time for the optimal activation of the intestinal immune system [13,26]. Alternatively, mice received *R. acadiensis* (10^9^ CFU/mL) for 30 or 90 days to analyze the innocuousness of the bacteria after long periods of administration. 

At the end of the *R. acadiensis* supplementation (7, 30 or 90 days), five mice of each control and test group were sacrificed by cervical dislocation and then small and large intestine, spleen and liver were removed for studies. Samples of intestinal fluids were also taken. All animal protocols were preapproved by the Animal Protection Committee of CERELA (CRL-BIOT-LI-2017/3C) and conducted in accordance with the guidelines established by the Consejo Nacional de Investigaciones Científicas y Técnicas (CONICET). 

### 2.3. Scanning and Transmission Electron Microscopy

This study was performed using BABL/c mice that received 100 µL of *R. acadiensis* (10^9^ CFU/mL) by gavage. One group of five (5) animals was sacrificed after 5 min and other groups (five mice) 15 min later. These times were established according to a previous report [27]. The small intestines of each mouse were removed, washed with 3 mL of PBS and 0.5 mm segments of tissue fixed in 2.66% formaldehyde, 1.66% glutaraldehyde, sodium phosphate buffer 0.1 M pH 7.4 and incubated overnight at 4 °C. The samples were processed by experts of the Centro Integral de Microscopía Electrónica (CIME-CONICET), and observed with a Zeiss EM109 (Carl Zeiss NTS GmbH, Oberkochen, Germany) and Zeiss SUPRA 55-VP for transmissions and scanning electron microscopy studies, respectively. 

### 2.4. Analysis of Some Total Populations of the Intestinal Microbiota

The large intestines were aseptically removed after consecutive 7, 30 and 90 days of oral *R. acadiensis* administration, weighed and placed into sterile tubes containing 5 mL of peptone water (0.1%). The samples were immediately homogenized under sterile conditions using a micro homogenizer (MSE, London, UK). Serial dilutions of the homogenized samples were obtained and aliquots (0.1 mL) of the appropriate dilutions were spread onto the surface of the different agarized media (Britania, Buenos Aires, Argentina); Mac Conkey (for enterobacteria); Man Rogosa Sharpe (MRS, for lactobacilli); Reinforced Clostridial agar (RCA, for total anaerobes). Mac Conkey and MRS agar plates were aerobically incubated at 37 °C for 24 h, while RCA plates were anaerobically incubated at 37 °C for 72–96 h.

### 2.5. Bacterial Translocation to Spleen and Liver

Animals that received a conventional diet or *R. acadiensis* (10^9^ CFU/mL) for 7, 30 and 90 days, were sacrificed at the corresponding times. The livers and spleens were aseptically removed, harvested in tubes containing 5 mL peptone water (0.1) and homogenized using a micro homogenizer (MSE, London, UK). One hundred microliters of the liver and spleen homogenate were spread onto the surface of MacConkey, MRS and TSA agar. The plates were aerobically incubated at 37 °C for 48 h. Translocation was considered to have occurred when colonies were observed on the agar plates [28].

### 2.6. Histology of Small Intestine

The small intestine of control animals and those fed for 7 and 90 days with *R. acadiensis* (10^9^ CFU/mL) were removed and small pieces of them were recovered and fixed in formaldehyde 10% solution in PBS pH 7. After fixation, the tissues were dehydrated and embedded in paraffin using conventional methods. 

Serial tissue sections of 4 µm from each paraffin block were made using a rotation microtome. Slices were stained with hematoxylin and eosin and analyzed in optical microscopy.

### 2.7. Immunohistochemically Analysis

After 7 days of feeding, the small intestine from experimental mice was removed for histological studies. Deparaffinized and rehydrated 5-μm tissue sections were rinsed with PBS, permeabilized with 0.1% Triton for 5 min and blocked for 10 min with 0.1% H_2_O_2_ to inhibit endogenous peroxidase activity. After rinsing in PBS, the sections were blocked with bovine serum albumin (1% BSA) for 30 min and then with normal goat serum (1/100).

The sections were incubated with the corresponding primary antibodies: E-cadherin antibody (sc-8426 Santa Cruz, Biotechnology) and occludin antibody (AA 480–520 antibodies-online Inc.) at 4 °C, overnight. Afterward, they were rinsed with PBS, and incubated at room temperature (RT) for 1 h with the secondary antibody peroxidase goat anti-rabbit IgG (Horseradish Peroxidase—HRP) (ab6721). Then the slices were incubated with a peroxidase (HRP) detection system (DAB Peroxidase Substrate. SK-4100 VECTOR Laboratories) at RT for 2–12 min and washed for 5 min in water. A counterstain was conducted with hematoxylin, before optical microscopy observation.

### 2.8. Intestinal Permeability

Groups of five mice fed by oral gavage with 10^9^ CFU/mL of *R. acadiensis* for 7 days received 0.6 mg/g body weight of fluorescein isothiocyanate (FITC)–labeled dextran (4kDa) (Sigma–Aldrich, Saint Louis, MO, USA) dissolved in PBS [29,30]. Mice were sacrificed 3.5 h later and blood was removed in the dark by cardiac puncture to measure the presence of FITC dextran in blood. Serum was separated by centrifugation and plasma FITC levels were determined using a fluorescence microplate reader (excitation 485 nm and emission 530 nm).

### 2.9. Intestinal Epithelial Cells Isolation and Cytokines Determination

The intestinal epithelial cells (IECs) from experimental animals fed for 7 or 90 consecutive days, were isolated from the small intestine according to Canali et al. [31]. Briefly, after bacteria administration, the animals were sacrificed and the small intestine was removed aseptically. The Peyer’s patches were discarded and the intestine was washed in PBS, incubated with PBS-dithiothreitol containing 0.03 M EDTA at 37 °C to eliminate residual mucus and shaken in RPMI 1640 (Sigma–Aldrich, Saint Louis, MO, USA) containing 0.01 M EDTA to disrupt the epithelium. IECs were washed, counted and adjusted to 1 × 10^6^ IECs cluster/mL. IECs suspensions were then transferred to six-well sterile plates (1 mL/well) and incubated for 18 h (37 °C/5% CO_2_). Supernatants were recovered for cytokine determination. Interleukin-6 (IL-6) and Interferon-gamma (IFN-γ) were determined using the corresponding enzyme-linked immunosorbent assay set according to the manufacturer’s instructions (BD OptEIA; BD Biosciences, San Diego, CA, USA).

### 2.10. Determination of Antimicrobial Activity from the Intestinal Fluid

The antimicrobial activity of the intestinal fluids of control animals and those fed with *R. acadiensis* was assayed according to Cazorla et al. [32]. Briefly, the small intestines of mice were removed and their content was collected in a sterile tube by passage of 0.5 mL of 10 mM sodium phosphate buffer, pH 7.4 along the intestine. The supernatant was then collected after centrifuging at 1300× *g* at 4 °C for 15 min. The exponential growth phase suspensions of *S*. Typhimurium and *S. aureus* adjusted at 5 × 10^8^ CFU/mL in 20 µL were incubated for 2 h at 37 °C in the presence of 100 µL of the intestinal fluids obtained from the different mice. Each incubation mixture was serially diluted, spread in duplicate selective agar plates, and incubated at 37 °C for 18 h, followed by the determination of CFU counts. Results were expressed as the CFU/mL of the pathogens after their incubation with the intestinal fluids.

Additionally, the antimicrobial peptide defensin 5 alpha, was measured in the intestinal fluids of the experimental mice by capture ELISA according to the manufacturer’s instructions (Defensin 5 Alpha, Paneth Cell Specific, ELISA Kit (#MBS2703886)).

### 2.11. Salmonella Typhimurium Infection

Groups of BALB/c mice weighing 26 ± 4 g were divided as follow: G-1: animals that received a conventional diet (Control); G-2: mice fed with a conventional diet and challenged by intragastric inoculation with 1 × 10^7^ CFU/mL of *Salmonella enterica* serovar Typhimurium (*Salmonella*-infected); G-3: animals fed with *R. acadiensis* (10^9^ CFU/mL) the 7 days previous to the Salmonella challenge (*R. acadiensis*-preventive) and G-4: animals fed with *R. acadiensis* (10^9^ CFU/mL) for 7 days prior to the Salmonella challenge, and uninterruptedly received *R. acadiensis* after the challenge. Animals were sacrificed 7 days post-infection. 

The liver and spleen were aseptically removed, weighed and placed into a sterile tube containing 5 mL of peptone water (0.1%). The samples were homogenized and serial dilutions were spread onto the surface of McConkey agar. The number of CFU was determined after aerobic incubation for 24 h at 37 °C. Results were expressed as CFU/g of an organ. 

### 2.12. Total and Specific Anti-Salmonella Secretory IgA (S-IgA)

The S-IgA antibodies in the intestinal fluid were measured by ELISA 7 days-post challenge. Anti-*Salmonella* IgA antibodies determination was carried out as described previously by Leblanc et al. [33] using goat anti-mouse IgA (alpha-chain-specific) conjugated peroxidase. The optical density was measured at 450 nm using a VERSA Max Microplate reader (Molecular devices, Sunnyvale, CA, USA). 

For the specific anti-*Salmonella* S-IgA antibodies determinations, plates were coated with 50 µL of a suspension of heat-inactivated *S.* Typhimurium (10^10^ CFU/mL) and incubated overnight at 4 °C. Nonspecific protein-binding sites were blocked with PBS containing 0.5% skim milk powder. The samples from the intestinal fluids of mice were diluted in 0.5% skim milk powder in PBS and then incubated at room temperature for 2 h. After washing with PBS containing 0.05% Tween 20, the plates were incubated for 1 h with peroxidase-conjugated anti-IgA-specific antibodies. Plates were again washed and the tetramethylbenzidine (TMB) reagent was added. The reaction was stopped with H_2_SO_4_ (2N). The absorbance was read at 450 nm. Results are expressed as concentration (µg/mL) of IgA in the intestinal fluid.

### 2.13. Statistical Analyses

Results are presented as means ± SEM. GraphPad Prism 5.0 software (GraphPad Software Inc., San Diego, CA, USA) was employed to carry out calculations. The results presented are representative of three independent experiments. No significant differences were observed between the three independent replicates. The statistical significance was determined by one-way analysis of variance (ANOVA), using Kruskal–Wallis test performed with the GraphPad Prism 5.0 software. Unless it was indicated, comparisons were referred to the mice that received a conventional balanced diet and water *ad libitum*. *p*-Values < 0.05 were considered significant.

## 3. Results

### 3.1. Effect of R. acadiensis Administration on Body Weight

The body weight of the mice fed with a conventional diet or the supplementation with 10^9^ CFU/mL of the bacterium for 30 or 90 consecutive days was determined every 2 or 4 days. We observed a slight increase in the body weight of mice supplemented with the bacterium for 30 days. However, no significant changes in body weight were observed in animals that received *R. acadiensis* for 90 days with respect to those that received a conventional diet (Figure 1). 

### 3.2. Evaluation of the In Vivo Adherence of R. acadiensis to the Intestinal Epithelium

*R. acadiensis* adherence to mice intestinal epithelial cells was analyzed on intestinal sections removed 5 and 15 min later the oral administration of the bacteria, by scanning and transmission electronic microscopy. We did not observe adhesion of the bacteria to the epithelial cell at any 5 or 15 min. Only the presence of the strain on the mucus layer was noted (Figure 2). 

### 3.3. R. acadiensis Oral Supplementation Did Not Disturb the Large Intestinal Homeostasis

To investigate the impact of *R. acadiensis* on the large intestinal microbiota, mice were fed with 10^9^ CFU/mL of the bacteria for 7, 30 or 90 consecutive days. *R. acadiensis* oral supplementation did not induce changes in the total lactobacilli and total anaerobe population at any-analyzed times. By contrast, the total enterobacterial population increased after *R. acadiensis* oral administration (Figure 3). 

### 3.4. R. acadiensis Reinforces the Intestinal Epithelial Barrier without Disturbing the Small Intestinal Homeostasis, Even after Oral Long-Term Consumption

Spleen and liver from animals supplemented with *R. acadiensis* during consecutive 7, 30, or 90 days were removed and plated on selective broth for enterobacteria and lactobacilli. After 24 h, no colonies were detected on McConkey or MRS agar plates; indicating no bacterial translocation of the intestinal microbiota to distant sites had taken place and the intestinal barrier had not been altered. 

Moreover, as shown in Figure 4A–D, a higher expression of cadherin and occludin proteins, components of tight junctions and adherens junction, respectively, were detected by immunohistochemistry studies in animals supplemented with *R. acadiensis* for 7 days compared with those that received a conventional diet (Figure 4A–D). To further address the effects of *R. acadiensis* on the intestinal epithelial barrier, permeability was determined in vivo using fluorescein-conjugated dextran as a tracer. Animals received FITC dextran by oral gavage the day after *R. acadiensis* supplementation, and 3.5 h later, FITC levels in the plasma were determined. No differences in the fluorescence levels were observed in animals supplemented with *R. acadiensis* with respect to those that received a conventional diet (2.21 ± 0.17 and 2.87 ± 0.29 µg/mL Plasma FITC, Mean ± SEM, respectively) (Figure 4E).

Additionally, we observed that *R. acadiensis* did not increase the release of cytokines after 7 or 90 days of consumption. Levels of IL-6 were 63.13 ± 16.83 and 64.32 ± 16.31 pg/mL, for 7 and 90 days of feeding, respectively. IFN-γ levels of 95.62 ± 0.28 and 99.76 ± 3.17 pg/mL, for 7 and 90 days of *R. acadiensis*-feeding, respectively, were observed. Controls presented values of IL-6 of 46.12 ± 0.60 and 44.32 ± 4.79 pg/mL and IFN-γ levels of 90.06 ± 2.17 and 123.10 ± 26.02 pg/mL at 7 and 90 days, respectively (Figure 5). 

We also demonstrated that even after long-term consumption (90 consecutive days) of *R. acadiensis* (10^9^ CFU/mL) the small intestine architecture of the animals was not altered. No inflammatory foci have been observed (Figure 6A). Interestingly, an increase in the number of Paneth cells, in *R. acadiensis*-fed animals, compared with animals that received a conventional diet (Figure 6B,C) was observed. Accordingly, high levels of the defensin 5-α, which limit the invasion and adherence of pathogenic and commensal bacteria, have been found in the intestinal fluids of mice supplemented by 7 days with the probiotic bacterium (*p* < 0.001) (Figure 6D).

### 3.5. Determination of Antimicrobial Activity from the Intestinal Fluid

In order to further characterize the antimicrobial activity of the intestinal fluids against microorganisms, samples of intestinal fluids from mice supplemented with *R. acadiensis* for either 7, 30 or 90 days were taken and assayed against pathogenic bacteria. An important decrease in the CFU/mL of *S*. Typhimurium and *S. aureus* was observed in the intestinal fluids of animals supplemented with *R. acadiensis* compared with the control group (Figure 7).

### 3.6. Continuous Administration of R. acadiensis Protects against Salmonella Typhimurium Infection

The increase in the number of Paneth cells and the in vitro antimicrobial activity of the intestine fluids of the animals supplemented with *R. acadiensis* led us to investigate in vivo whether the probiotic candidate protects against *S.* Typhimurium infection. Animals that received a conventional diet (Control) or the supplementation with *R. acadiensis* for 7 days were infected with 1 × 10^7^ CFU/mL of *Salmonella enterica* serovar Typhimurium. An additional group of mice fed with *R. acadiensis* still received the probiotic bacteria for an additional 7 days after the challenge (continuous feeding). Interestingly, we observed that these mice showed a better survival capability for *Salmonella* infection than those that received a conventional diet (Figure 8A). Additionally, when we analyzed the translocation of the pathogen bacteria to both the liver and spleen, we observed a decrease in CFU/mL in animals that received a continuous supplementation with *R. acadiensis* compared with infected animals that received a conventional diet, (*p* < 0.05) (Figure 8B). These results suggested that oral administration of *R. acadiensis* reinforced the intestinal epithelial barrier protecting against *Salmonella* dissemination. By contrast, the consumption of *R. acadiensis* before *Salmonella* infection (*R. acadiensis*-preventive) was not able to protect against the Salmonella spread to different tissues. 

Finally, levels of intestinal IgA were analyzed in the *Salmonella*-infected animals. We observed that total and specific anti-*Salmonella* s-IgA did not increase upon preventive or continuous administration of *R. acadiensis*, compared with infected mice receiving a conventional diet. Interestingly, total s-IgA in infected animals that received continuous *R. acadiensis* administration were similar to those observed in uninfected control mice (9.798 ± 1.246, and 11.69 ± 1.148, respectively) (Figure 8C). Additionally, the DO to 450 nm that represented the specific anti-*Salmonella*-s-IgA was 0.62 ± 0.056 and 0.65 ± 0.097, for *Salmonella*-infected and *R. acadiensis* infected and uninterruptedly supplemented animals, respectively (Figure 8D). Altogether, these results suggest that *R. acadiensis* had a protective effect against *Salmonella* infection by avoiding systemic dissemination.

## 4. Discussion

Probiotics are currently used for prophylaxis and therapy in several diseases including antibiotic-associated diarrhea, infectious childhood diarrhea, ulcerative colitis, pouchitis or atopic eczema associated with cow milk allergy [34]. The therapeutic use of probiotics for other diseases such as metabolic syndrome and type 2 diabetes mellitus, or the prophylactic use for infections and allergies, has been previously studied [35,36,37,38]. 

Most of the commercially available probiotics belong to a limited list of genera, including *Lactobacillus* spp. and *Bifidobacterium* spp. [39]. The most commonly exploited strains/species among Lactobacilli and Bifidobacteria have been granted generally recognized as safe status by the U.S. Food and Drug Administration (FDA, 2021) [40]. 

In the present study, we analyzed the probiotic potential of *R. acadiensis*, by reporting its effect on the maintenance of gut barrier as well as its anti-microbial effect. *R. acadiensis* differs from the strains most used as probiotics since most of them belong to the genus *Lactobacillus* or *Bifidobacterium*. Of note, Gram (-) probiotic bacteria such as *Escherichia coli Nissle 1917* have been largely studied over the years. This new potential probiotic strain has been demonstrated by genomic prediction completed by the National Research Council of Canada (NRC) and by in vitro studies, not to show virulence genes nor resistance to antibiotics [24]. The authors also demonstrated its ability to resist the adverse conditions encountered during its passage through the intestinal tract following oral administration [23]. In in vitro preliminary studies, no hemolytic nor cytotoxic activity of the bacterium was observed. This prompts us to further investigate *R. acadiensis* potential as a probiotic bacterium and its anti-microbial effects against *S*. Thyphimurium infection, a common intestinal pathogen. 

Interestingly, we observed an increase in body weight in the groups of mice that received *R. acadiensis* for 30 days; however, this increase did not represent an alarming percentage with respect to the control. When we followed the increase in body weight in the group of mice that received the bacteria for a longer period of time (90 days), we observed that the kinetics were comparable to those of the control group receiving a conventional diet (Figure 1), providing additional evidence of its safety as a potential probiotic bacterium. In same line of observation, the intestinal architecture was shown to be not affected by short- or long-term administration of *R. acadiensis* (7, 30 or 90 days). Although it has been reported an adhesion rate to *R. acadiensis* nearing 20% on Caco-2 and HT29 cell lines [23], these results were not confirmed by in vivo studies. Scanning and transmission electronic microscopy performed in mice orally supplemented with *R. acadiensis,* showed that the new probiotic bacterium was unable to adhere directly to the intestinal epithelial cells by remaining in the mucus layer (Figure 2). 

The role of the gut microbiota in health and disease and their influence on the immune system is well established [41,42]. The disruption in the equilibrium of the gut microbiota can induce inflammatory diseases and metabolic disorders. The total bacteria populations of the large intestine were not significantly modified in *R. acadiensis*-supplemented animals, compared with those that received a conventional diet (Figure 3), neither after short- or long-term administration. A slight increase in the Enterobacteriaceae family was observed. Noteworthy, after 90 days of supplementation with *R. acadiensis*, an increase lactobacilli and a decrease in enterobacteria was noted. These results suggest that *R. acadiensis* increased lactobacilli which are considered beneficial microorganisms. Ongoing metagenomics analysis, after *R. acadiensis* oral administration will be able to corroborate this effect. 

Bacteria translocation is the migration of microorganisms present in the intestinal lumen across the epithelium and its dissemination to extra-intestinal sites such as the liver and spleen [43]. In the present study, we found that oral supplementation of *R. acadiensis* did not cause the translocation of bacteria from the intestinal microbiota to the liver and spleen. In addition, changes in intestinal epithelial permeability by the oral intake of *R. acadiensis* supplementation were also observed following FITC–labeled dextran administration (Figure 4E), thus confirming that supplementation by *R. acadiensis* is neither affecting the gut barrier nor causing translocation to distant tissues. 

Probiotics exhibit the potential to maintain intestinal homeostasis and prevent bacterial translocation via enhancing intestinal barrier function [44]. The intestinal physical barrier is composed of the commensal microbiota, the mucus layer and the intestinal epithelial layer [45]. A central element of the mammalian intestinal to maintain gut homeostasis is to minimize contact between luminal microorganisms and the intestinal epithelial cell surface. This is accomplished by the production of mucus, antimicrobial proteins and secretory IgA [46,47]. The intestinal epithelial cells are joined by a tight junction to form a contiguous and relatively impermeable barrier, inhibiting the translocation of pathogens and bacteria from the gut to distant tissues. When we analyzed the presence of proteins that are components of the tight junctions (TJ) and adherens junction (AJ), we observed that cadherin and occludin were expressed in higher levels in the intestinal epithelial cells of *R. acadiensis*-fed animals compared with animals fed a conventional diet (Figure 4).

Moreover, an increase in the number of Goblet and Paneth cells in the intestines of *R. acadiensis*-supplemented animals by the microscopic observation of stained small intestine tissues was observed. Based on these results, we decided to investigate the release of one of the main antimicrobial peptides secreted by the Paneth cells to the lumen. Accordingly, high levels of defensin 5-α in the intestinal fluids of these mice were detected (Figure 6). Several reports highlighted the important role played by Paneth cells through the secretion of antimicrobial peptides, in the protection against pathogen colonization, the strengthening of the intestinal barrier integrity, and the control of the microbiota composition and localization [48,49]. 

The experiment pertaining to the anti-microbial effects against *S*. Typhimurium and *S. aureus*, showed a decrease in the number of CFU/mL of both bacteria. This was noted on plate count agar after the incubation of the pathogens in the presence of the intestinal fluids of mice fed for 7, 30 or 90 days with *R. acadiensis* (Figure 7). 

The intestinal epithelial cells are responsible for coordinating the mucosal immune response by releasing chemokines and cytokines that recruit immune cells from both the innate and adaptive immune response. Signals generated by the resident microbiota on the IECs are crucial to maintaining the homeostasis of the immune system. Therefore, the cytokine release by the intestinal epithelial cells following *R. acadiensis* administration in an ex vivo assay was analyzed. No increase in inflammatory cytokines (IL-6 and IFN-γ) was reported in animals that received *R. acadiensis* for 90 consecutive days compared with the control (Figure 5). In a previous study, it was demonstrated that *R. acadiensis* increased the number of IL-10+ cells in the lamina propria and also the secreted IL-10 in the intestinal fluid, suggesting their anti-inflammatory and regulatory properties [23]. Infections of the gastrointestinal tract represent a major global health problem, with Salmonella serotypes and enterohemorrhagic *E. coli* being the most common microorganisms involved. *Salmonella* Typhimurium is an invasive bacterium, which through the M cell of the Peyer’s patches, invades the immune cells associated with the gut and disseminates toward deep tissues causing diarrhea. The integrity of the epithelial barrier is crucial to protect against Salmonella infection and prevent its dissemination to distal tissues. Secretory IgA also plays an important role in controlling infection and preventing the dissemination of pathogens. Some probiotic strains have been shown to protect against Salmonella infection [50,51,52]. We explored whether *R. acadiensis* could protect against a *Salmonella* Typhimurium infection. A significant decrease in the dissemination of Salmonella to liver and spleen and in the mortality of the mice was achieved after a continuous supplementation with *R. acadiensis* (Figure 8). The underlining protective effects of *R. acadiensis* against *Salmonella* Typhimurium might be linked to its effect on the reinforcement of the intestinal barrier, mediated by tight junctions and the secretion of antimicrobial peptides. 

S-IgA immunoglobulin levels to a *Salmonella* Typhimurium infection were similar in R. acadiensis-fed animals compared with those that received a conventional diet. The supplementation with *R. acadiensis* prevented the anti-*Salmonella*-s-IgA increase indirectly linked to its capacity to control the infection and prevent the pathogen dissemination.

## 5. Conclusions

*R. acadiensis* reinforced the intestinal epithelial barrier by increasing the expression of the tight junction proteins, the intestinal antimicrobial activity and the antimicrobial peptides production. This effect was elicited without generating an inflammatory immune environment since neither IL-6 nor IFN-γ proinflammatory cytokines increase, even after long-term continuous supplementation (90 days) at a daily dose of 10^9^ CFU/mL. Future clinical studies and investigations will warrant *R. acadiensis* to be considered as a novel next-generation probiotic bacterium. 

## Figures and Tables

**Figure 1 microorganisms-11-01347-f001:**
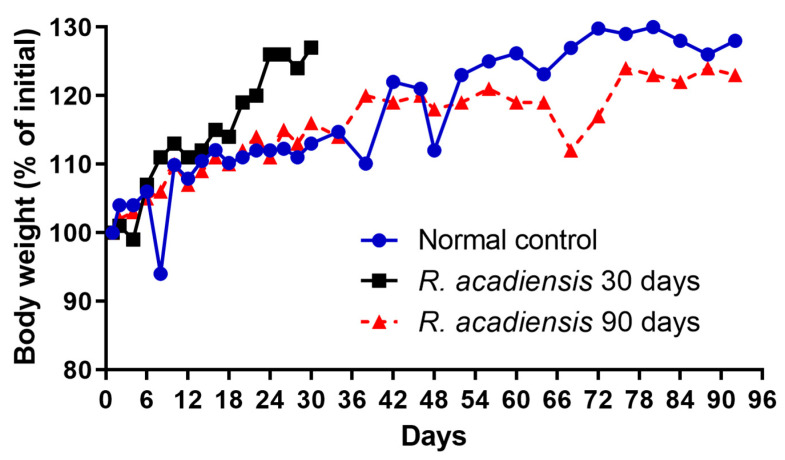
Influence of *R. acadiensis* oral supplementation on the body weight. Animals were fed with conventional diet (control) or 10^9^ CFU/mL of *R. acadiensis* for 30 or 90 days. The body weight was determined every 2 or 4 days. Results were expressed as the % of mean of the initial weight (weight registered the day before bacteria administration).

**Figure 2 microorganisms-11-01347-f002:**
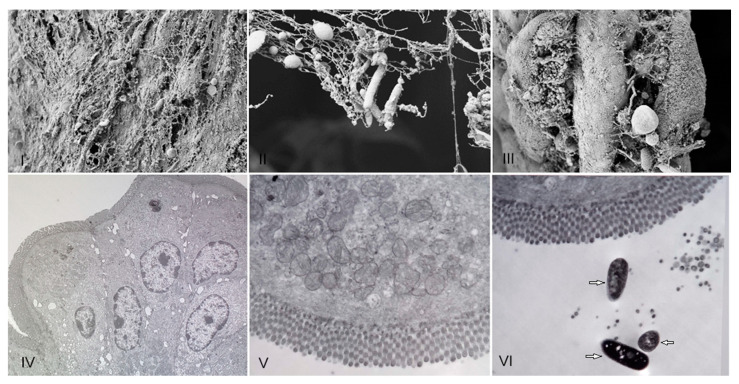
Presence of *R. acadiensis* on the mucus intestinal layer. Water (control (**I**,**IV**)), or *R. acadiensis* (10^9^ CFU/mL, (**II**,**III**,**V**,**VI**)) were administered by intragastric intubation to BALB/c mice. Five (Panels: (**II**,**V**)) or 15 min later (Panels (**III**,**VI**)), mice were killed and their small intestine were removed, washed with PBS, fixed and incubated overnight at 4 °C. The samples were processed for scanning electron microscopy (**I**–**III**), and transmission electron microscopy (**IV**–**VI**). Arrows indicate bacilli presence on the intestinal mucus. Magnification: (**I**)—10.00 K×; (**II**)—29.79 K×, (**III**)—9.09 K×. (**IV**)—4.85 K×, (**V**)—7.05 K×, (**VI**)—7.05 K×.

**Figure 3 microorganisms-11-01347-f003:**
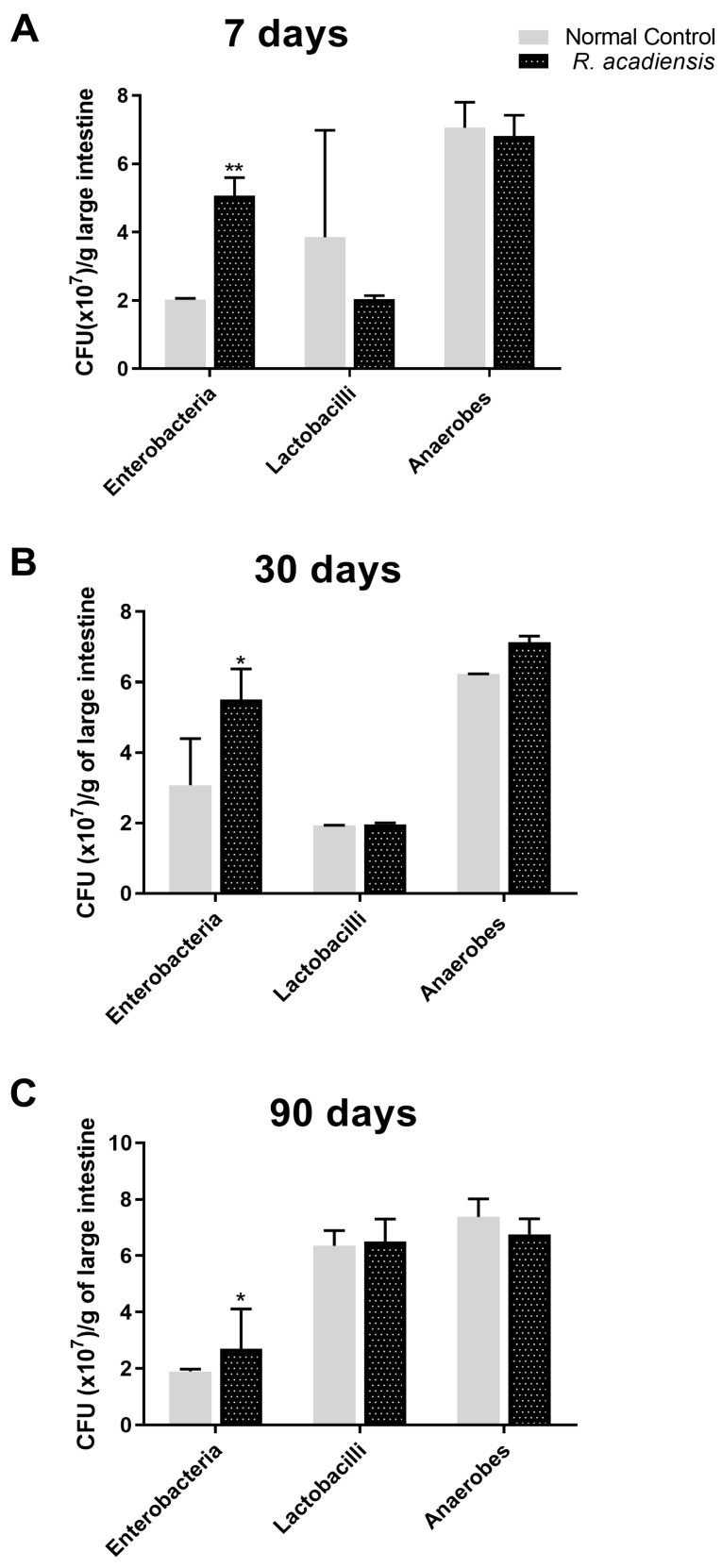
Influence of R. acadiensis on large intestine microbiota. Mice were fed with conventional diet (Control) or 10^9^ CFU/mL of *R. acadiensis* for consecutive (**A**) seven days, (**B**) 30 days or (**C**) 90 days, respectively. At the end of these times, samples of large intestine were collected and total anaerobic bacteria, lactobacilli, and enterobacteria populations were analyzed by plate count agar. Results were expressed as CFU/mL per gram of large intestine (mean ± S.E.M). Three independent experiments were performed. * *p* < 0.05, ** *p* < 0.01.

**Figure 4 microorganisms-11-01347-f004:**
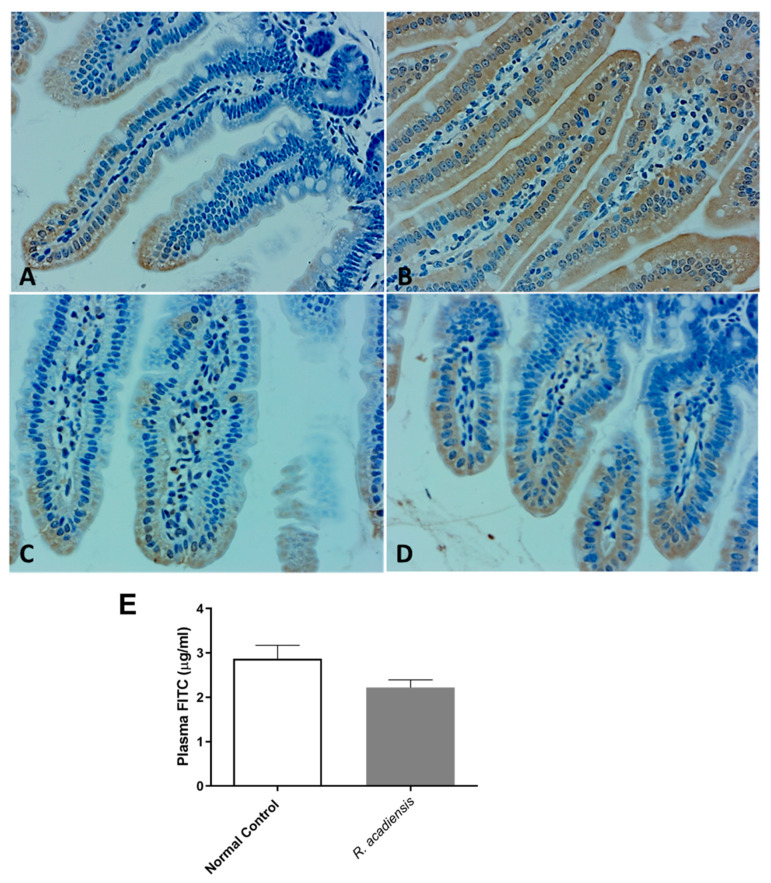
Intestinal epithelial barrier in mice fed with conventional diet (Normal Control) or *R. acadiensis* for 7 days. At the end of this time mice were sacrificed and sections of small intestine were taken for immunohistochemistry of the expression of cadherin (**A**,**B**) and occludin (**C**,**D**). Panel (**A**,**C**): Normal Control; Panel (**B**,**D**): *R. acadiensis* fed animals. Magnification ×400. (**E**) Fluorescence measured in the blood of the animals at 3.5 h later FITC dextran administration. Mice were previously fed with a conventional diet (Normal Control) or *R. acadiensis* for 7 days. Results are expressed as the Mean ± SEM of three independent experiments.

**Figure 5 microorganisms-11-01347-f005:**
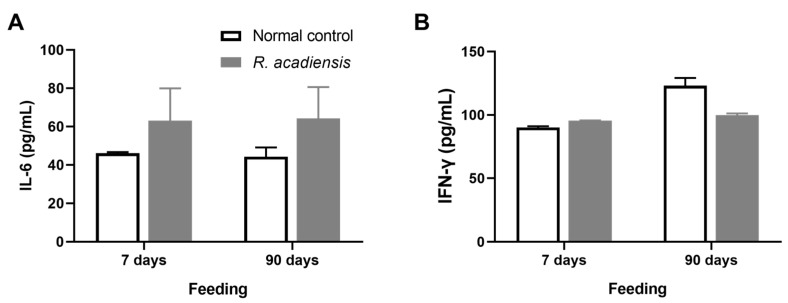
Cytokines release by intestinal epithelial cells. IL-6 (**A**) and IFN-γ (**B**) were determined in the supernatant of epithelial cell culture from mice fed upon a conventional diet or the supplementation with or *R. acadiensis* (10^9^ CFU/mL) for 7 or 90 days. Samples were assayed in duplicate by capture ELISA. Results are expressed as the Mean ± SEM.

**Figure 6 microorganisms-11-01347-f006:**
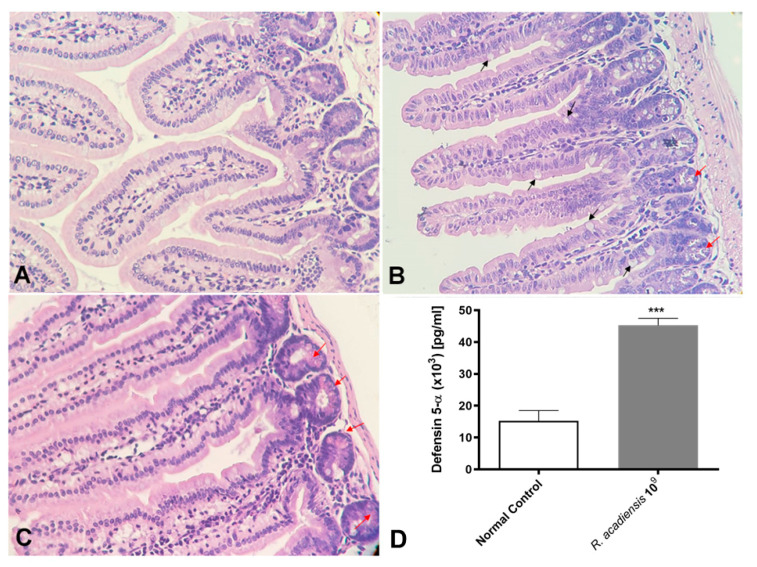
Histological studies of the intestinal epithelial tissue. Animals were fed with (**A**) conventional diet (control) or (**B**,**C**) *R. acadiensis* (10^9^ CFU/mL) for 7 days. At the end of these times small intestines were removed and tissue sections were stained with hematoxylin and eosin. Red arrows indicate Paneth cells, while black arrows denote Goblet cells. Magnification ×400. (**D**) the defensing 5-α antimicrobial peptide secreted by Paneth cells was determined by ELISA in the intestinal fluids of animals fed with conventional diet (Normal Control) or 10^9^ CFU/mL of *R. acadiensis.* Results are expressed as the Mean ± SEM. *** *p* < 0.001.

**Figure 7 microorganisms-11-01347-f007:**
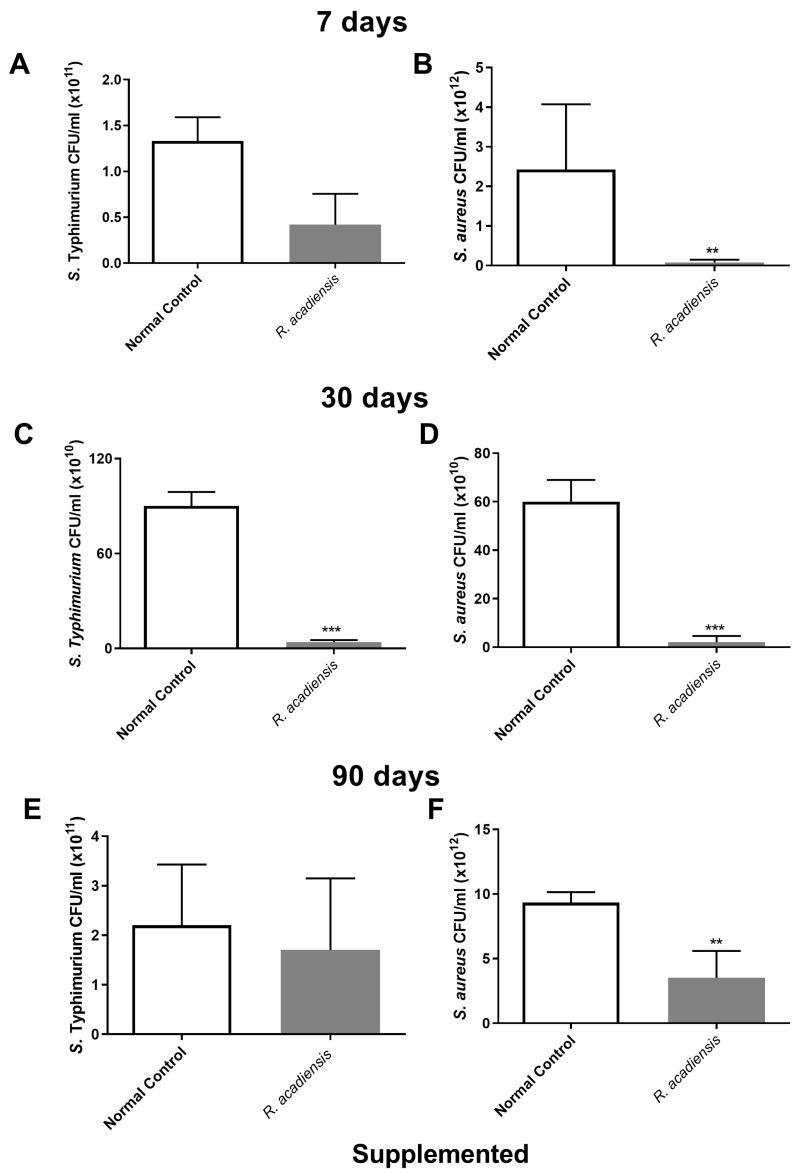
Determination of the antimicrobial activity in animals’ intestinal fluids. *S. aureus* and *S.* Typhimurium (5 × 10^10^ CFU/mL) were incubated for 2 h at 37 °C in the presence of the intestinal fluids of mice fed with a conventional diet (Normal control), or *R. acadiensis* for consecutive 7 (**A**,**B**), 30 (**C**,**D**) or 90 (**E**,**F**) days. After the co-incubation, viable bacteria were determined by plate counts. Results are expressed as CFU/mL of three independent experiments. ** *p* < 0.01, *** *p* < 0.001.

**Figure 8 microorganisms-11-01347-f008:**
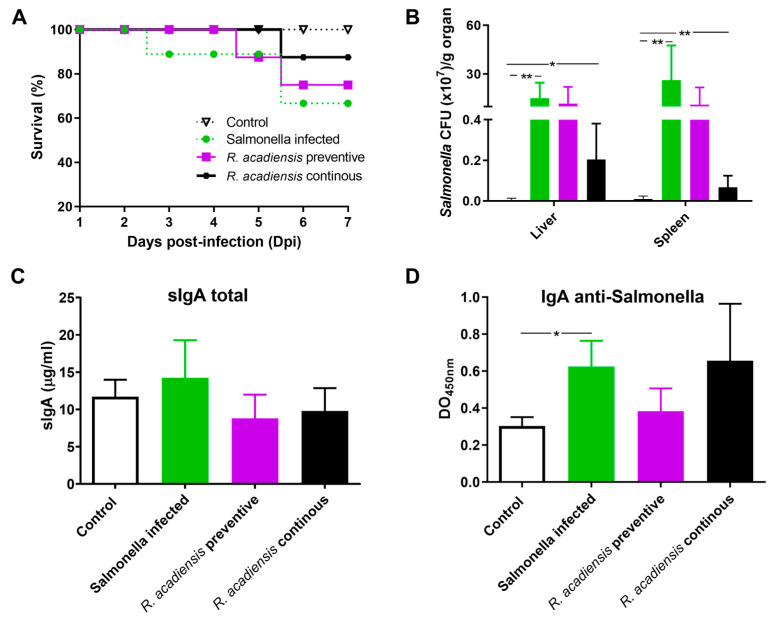
Challenge with *S*. Typhimurium in *R. acadiensis*-fed mice. Mice fed with a conventional diet or *R. acadiensis* for 7 days were orally infected with *S*. Typhimurium. After the challenge, one additional group of mice was uninterruptedly supplemented with *R. acadiensis* for 7 days more (*R. acadiensis*-continuous). (**A**) Survival of the animals to a *Salmonella* oral infection. (**B**) Translocation of the pathogen bacteria to liver and spleen in mice at day 7-post infection. Levels of total sIgA (**C**) and specific anti-*Salmonella* IgA (**D**) determined by ELISA on blood samples taken at the end of the experiments. Results are expressed as the Mean ± SEM of three independent experiments. * *p* < 0.05, ** *p* < 0.01.

## Data Availability

Not applicable.

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
