# Peer review of "Evaluation of *Rouxiella badensis* Subsp *Acadiensis* (Canan SV-53) as a Potential Probiotic Bacterium"

_microorganisms, 2023, doi:10.3390/microorganisms11051347_

Round 1

Reviewer 1 Report

Dear Authors,

Although interesting, from the point of potentially including and defining novel probiotic strains, the manuscript suffers from a number of shortcomings. 

First of all, the language is quite challenging to read, there are numerous misspelled words and grammar mistakes (too many to list, but some are seriously misleading, like for example mentioning intestinal bowel diseases (IBD) on line 75, where one would expect Inflammatory Bowel Diseases). 

The title is also misleading since the activity against pathogens is not described in detail until the last part of the manuscript, in which a protective effect is being mentioned against S. typhimurium infection.

Another important issue is the statistics, nowhere did I find the number of animals used in the study/group mentioned and based on that no conclusion can be made on the power of the study. 

My advice would be to seriously edit the existing manuscript, change the title, perhaps include 16S rRNA amplicon sequencing in order to really assess the microbiota, since the homestasis is being frequently mentioned. The focus of the revised manuscript should not be on the protective effect since it lacks evidence, but on "the selection of an appropriate probiotic microorganism, that should be made on the basis of its capacity to improve the gut ecosystem without disrupting the intestinal homeostasis" as mentioned early in the text, which is much more in line with the work being described.

Unfortunately, the language of the manuscript is seriously flawed and I would highly recommend some external editorial services for the authors to help them produce a better manuscript. At times, it was very difficult to read and some typos are even misleading (like mentioning Intestinal Bowel Disease (IBD), where it is not clear if this is some new syndrome or a reference to Inflammatory Bowel Disease).

Reviewer 2 Report

REVIEW

Dear authors,

Please consider the following comments to improve the content of your manuscript before publication. 

The work proposes the use of a Gram (-) bacterium, isolated from a non-human source as a potential probiotic. To date, only one commercial probiotic belongs to the Proteobacteria (Escherichia coli Nissle 1917), so presenting evidence of the safety and beneficial effects of Rouxiella acadiensis in an in vivo model opens the possibility of its use in the future.

As the authors mention, probiotics must demonstrate a series of functional characteristics such as adherence, competition with pathogenic microorganisms, low immunogenicity (immunomodulation) and more recently, they do not cause an imbalance in the intestinal microbiota (dysbiosis). Despite the fact that the work shows evidence of each of these characteristics, I find a great methodological deficiency, since at no time is the TOTAL number of mice used, as well as their approximate age (weeks), mentioned; they must specify the number of mice for each of the experiments performed.

In Figure 3 you mention that 28 week old BALB/c mice were used, did you use mice of this age to perform this particular experiment or was that the age of the mice in all the experiments?

Why did they use mice of this age (28 weeks) to assess the culturable microbiota in the large intestine?

What is the explanation that the CFU of Enterobacteria decrease after 90 inocula compared to treatment with 7 and 30 inocula of R. acadiensis?

Could R. acadiensis be modifying the microenvironment so that other commensal Enterobacteriaceae increase (abundance)?

What role do lactobacilli play after 90 consecutive inocula with R. acadiensis?

Lactobacilli immunoregulate prolonged contact with antigens from Gram (-)?

They could evaluate the systemic immune response through serum, since it is not only immunomodulated at the in situ level (intestine), probiotics also have the ability to regulate at the organism level (systemic), in addition, it would be necessary to add an anti-inflammatory cytokine such as the IL-10.

Staphylococcus aureus and Salmonella Typhimurium strains are ATCC?

It is necessary to make the following corrections in the indicated lines:

Lines 2, 72, 73, 232, 246, 247, 248, 257, 259, 263, 264, 311, 321, 391, 420, 426, 427, 428, 429, 430, 435, 436, 438, 442, 443, 446, 447, 448, 449, 450, 464, 470, 471, 534, 546, 550, 552, 553, 555, 563: the scientific names, as well as their abbreviation, must be written in italics.

Lines 25, 87, 122, 281, 301, 345, 346, 358, 413, 414, 415, 474, 476, 489, 542: the terms in vivo, in vitro, ex vivo and ad libitum must be written in italics.

Line 19: remove commas and put parentheses “, R. acadiensis,

Line 22: homogenize mouse strain name “BALB/c”.

Line 24: write the full name of the genera of microorganisms “Staphylococcus aureus” y “Salmonella enterica serovar Typhimurium”.

Lines 24, 25: write the correct name of the species “Typhimurium”.

Line 104: remove the apostrophe in “R. acadiensis”.

Lines 110, 119, 147, 158, 170, 180, 189, 204, 212, 226, 242, 257, 274: place the numbering corresponding to each method (2.1, 2.2, 2.3…2.13).

Lines 122, 176: the degree centigrade symbol is smaller than its superscript “°C”.

Lines 148, 151, 173, 230, 233, 236, 271, 332, 379, 402: write the units of volume correctly “mL” y “µL”.

Line 192: write in subscript “H2O2”.

Line 221: place the symbol diagonally and write in subscript “37°C/CO2”.

Line 222: write the full name of the cytokines before abbreviating them “Interleukin-6 (IL-6) and Interferon-gamma (IFN-γ)”.

Line 239: write in capital letters “ELISA”.

Figure 1: change the color of the group “R. acadiensis 30 days”, is not observed in the graph. Why does the body weight of the mice with 30 inocula increase compared to the 90-day mice in the period of 18-30 days?

Figure 2: write the text since the explanation of the Figure is not understood, the arrows that indicate the adherence of the bacilli to the mucosa are not observed.

Figure 3: correct the unit of grams on the Y axis “/g of large intestine”.

Figure 4: indicate the control group and R. acadiensis. Add measure bar and target magnification.

Table 1: I believe that these results should be presented in graphical form, they do not indicate whether there are significant differences between the values of each cytokine for each treatment.

Lines 374, 375, 376: the text is part of the title of Table 1?

Figure 5: missing measure bar and target magnification.

Lines 407, 572: write in superscript “109”.

Lines 408, 452: write in lowercase “p<0.05”.

Figure 6A: it is not indicated what “Dpi” means, write the full text “Days post-infection”. Change the color of the "Salmonella infected" group, it is not distinguished in the graph.

Figures 6B and 6D: the significant differences with respect to which group are they?

Line 450: indicate the name of R. acadiensis instead of “S. vaccinii”.

Line 471: write full name “Escherichia coli Nissle 1917”.

References

Some of the references must be removed from the journal number, the volume and the scientific names must be written in italics.

Please amend the requested comments and submit the revision file.

The text is easy to understand, has an appropriate way of developing the discussion.

Reviewer 3 Report

Manuscript titled “Rouxiella badensis subsp. acadiensis (Canan SV-53) as a poten- 2 tial probiotic bacterium against pathogens ” represents an interesting problem.

The more so that it is a continuation of the two work from 2021: „Immunomodulation and Intestinal Morpho-Functional Aspects of a Novel Gram-Negative Bacterium Rouxiella badensis subsp. acadiensis” (doi: 10.3389/fmicb.2021.569119) (Literature numer – 27) and „Adolescent use of potential novel probiotic Rouxiella badensis subsp. acadiensis (Canan SV-53) mitigates pubertal LPS-Induced behavioral changes in adulthood in a sex-specific manner by modulating 5HT1A receptors expression in specific brain areas” (doi: 10.1016/j.cpnec.2021.100063).

However, please pay attention to the nomenclature of microbes. Correct nomenclature is very important in microbiological research. Without it, the publication may lose its validity, it cannot be ignored!
